# Protective Effect of Natural Antioxidant, Curcumin Nanoparticles, and Zinc Oxide Nanoparticles against Type 2 Diabetes-Promoted Hippocampal Neurotoxicity in Rats

**DOI:** 10.3390/pharmaceutics13111937

**Published:** 2021-11-16

**Authors:** Shaymaa Abdulmalek, Mayada Nasef, Doaa Awad, Mahmoud Balbaa

**Affiliations:** 1Department of Biochemistry, Faculty of Science, Alexandria University, Alexandria 21511, Egypt; shimaa_salamy@yahoo.com (S.A.); mayadanasef90@gmail.com (M.N.); doaaelsayed363@hotmail.com (D.A.); 2Center of Excellency for Preclinical Study (CE-PCS), Pharmaceutical and Fermentation Industries Development Centre, City of Scientific Research and Technological Applications (SRTA-City), New Borg El-Arab City 21934, Egypt

**Keywords:** diabetes mellitus, neurotoxicity, curcumin nanoparticle, zinc oxide nanoparticle, metformin, inflammation, oxidative stress, apoptosis

## Abstract

Numerous epidemiological findings have repeatedly established associations between Type 2 Diabetes Mellitus (T2DM) and Alzheimer’s disease. Targeting different pathways in the brain with T2DM-therapy offers a novel and appealing strategy to treat diabetes-related neuronal alterations. Therefore, here we investigated the capability of a natural compound, curcumin nanoparticle (CurNP), and a biomedical metal, zinc oxide nanoparticle (ZnONP), to alleviate hippocampal modifications in T2DM-induced rats. The diabetes model was induced in male Wistar rats by feeding a high-fat diet (HFD) for eight weeks followed by intraperitoneal injection of streptozotocin (STZ). Then model groups were treated orally with curcumin, zinc sulfate, two doses of CurNP and ZnONP, as well as metformin, for six weeks. HFD/STZ-induced rats exhibited numerous biochemical and molecular changes besides behavioral impairment. Compared with model rats, CurNP and ZnONP boosted learning and memory function, improved redox and inflammation status, lowered Bax, and upregulated Bcl2 expressions in the hippocampus. In addition, the phosphorylation level of the MAPK/ERK pathway was downregulated significantly. The expression of amyloidogenic-related genes and amyloid-beta accumulation, along with tau hyperphosphorylation, were lessened considerably. In addition, both nanoparticles significantly improved histological lesions in the hippocampus. Based on our findings, CurNP and ZnONP appear to be potential neuroprotective agents to mitigate diabetic complications-associated hippocampal toxicity.

## 1. Introduction

Type 2 diabetes mellitus (T2DM) is associated with an increased risk of neurocognitive dysfunction and Alzheimer’s disease (AD). As a progressive and irreversible disease, AD is characterized by neuronal cell death, increased neurofibrillary tangles deposition inside cells, and amyloid plaque production in the gaps between neurons [1]. There is currently no viable treatment for this complex condition [2]. Additionally, in AD, tau undergoes various post-translational changes in addition to phosphorylation, which are thought to be involved in its pathological assembly, causing the breakdown of neuronal cells [3].

While the etiology of AD remains unknown, numerous studies increasingly focus on the risk factors that may shed light on the pathophysiology of the disease, such as overnutrition, aging, and T2DM [4]. Amongst various risk factors, obesity, a public health problem in Western nations, is thought to be the prominent link between T2DM and neurocognitive impairment. Furthermore, obese animal models have shown impaired insulin signaling before developing T2DM [5]. It has also been documented those changes in carbohydrate metabolism play an essential role in the pathophysiology of T2DM and act as an important contributor to the pathogenic mechanisms related to reduced cognitive performance [6,7].

Moreover, it is worth noting that chronic inflammation is a common risk factor for developing T2DM and AD. Many dementia-related neurodegenerative disorders are linked to the start and progression of neuroinflammation [8]. It is characterized by enhanced activation of microglia and is frequently accompanied by an increase in inflammatory cytokines, such as interleukin-1 (IL-1) and tumor necrosis factor (TNF) [9]. Various investigations have found that neuroinflammatory alterations develop with age, even in healthy brains, and include increased levels of cytokines in serum and CSF [10,11]. These accumulated alterations can be cytotoxic and affect critical neural functions, resulting in the development of neurodegenerative diseases [12].

Among several potential molecular pathways, the MAPK/ERK pathway is an evolutionarily well-preserved signaling mode that regulates fundamental cellular processes, such as proliferation, cellular survival, and differentiation. In addition, it provides an excellent example of such kinase cascades and as a therapeutic target for neurodegeneration [13,14]. There are three protein families: extracellular signal kinase (ERK), p38 kinase, and c-Jun N-terminal kinase (JNK) families. ERK1/2 and p38 MAPK are triggered under different cellular processes, including stress and inflammatory conditions, and regulate the expression of pro-inflammatory cytokines [15]. They also modulate the apoptotic pathway [16]. Moreover, Amyloid-beta (Aβ) can modify the ERK/MAPK pathway and trigger memory impairment [17].

Considering the biochemical link between AD and T2DM [18], a similar therapeutic possibility may exist for both diseases. Due to their effectiveness, relatively few side effects, and simple accessibility, natural compounds may provide an alternative medication for diabetes and the promise of a new therapy for AD [19]. Historically, natural medicinal plants have played and will continue to play an important role in drug discovery as they possess valuable therapeutic properties. The isolated bioactive compounds derived from herbal plants can be used for the discovery of new drugs. Curcumin is one of the natural phytochemicals in turmeric and is a potent anti-amyloidogenic agent which has been found to effectively disaggregate Aβ species directly, which in turn hinders the formation of fibrils and oligomers and improves cognitive performance [20,21]. In addition, curcumin attenuated the oxidative stress in a streptozotocin (STZ)-induced model by enhancing reduced glutathione (GSH) content and reducing the acetylcholinesterase activity in the hippocampus and cerebral cortex [22,23]. Previous studies showed that curcumin increased lifespan and decreased amyloid neurotoxicity in an invertebrate model [24,25].

Unfortunately, some studies revealed that pure curcumin has numerous toxic effects on living tissues [26,27]. In addition to this, the insufficient absorption and low bioavailability of pure curcumin are the major problems when it comes to oral administration [28]. A possible solution to these problems would be to develop curcumin nanoparticles (CurNP) to enhance their stability, bioavailability, and reduce the undesirable effects [29].

Recently, biomedical nanomaterials have received more attention because of their salient biological features and biomedical applications. With the development of nanomaterials, metal oxide nanoparticles exhibit promising and far-ranging prospects in biomedicine [30]. Zinc oxide nanoparticles (ZnONP)—one of the most important classes of metal oxide nanoparticles—are commonly utilized in various fields due to their physical and chemical properties [31]. ZnONP, a new type of low-toxicity and low-cost nanomaterial, has attracted great interest in different biomedical fields, including anticancer, antioxidant, anti-inflammatory, and antidiabetic activities, as well as for bioimaging applications and drug delivery [32,33].

Although studies are implicating the adverse effects of ZnONP on the animal; and indicating its harmful effect on oxidative stress and inflammation, not enough studies are being carried out to investigate the neuroprotective actions of ZnONP in obese-T2DM animal models. However, the capability of ZnONP in mitigating diabetes-induced neurotoxicity in the hippocampus and improving diabetes-associated cognitive changes compared to its classical form, zinc sulfate, is still incomplete and partially contradictory.

Through these biological effects of CurNP and ZnONP, the study reported here focused on the therapeutic efficacy of prepared, naturally occurring CurNP and a biomedical metal oxide, ZnONP, compared with their classical forms, curcumin and zinc sulfate, as well as an antidiabetic drug, metformin, on learning and memory functions, the main amyloidogenic cascade, neuroinflammation, oxidative stress, and neuro-apoptosis caused by a high-fat diet (HFD) and STZ administration in the hippocampus of rats. We also examined the role of prepared nanoparticles on the MAPK/ERK pathway in the hippocampus of T2DM-induced rats to figure out the most effective and safe compound(s) among the current therapies and find the effective dose, and to support the rationale for using naturally occurring nanoparticles and metal oxide-based nanomaterials in clinical trials for preventing or treating T2DM and its complications, particularly AD.

## 2. Materials and Methods

### 2.1. Preparation and Characterization of CurNP and ZnONP Nanoparticles

Curcumin nanoparticles were prepared using a dropwise method. Briefly, pure curcumin powder (Sigma-Aldrich, St. Louis, MO, USA) was dissolved in Dichloromethane (Sigma-Aldrich, MO, USA) to prepare a stock solution, 5 mg/mL. Then 1 mL from curcumin stock solution was added to 50 mL boiling water under ultrasonication for around 30 min, then stirred for 20 min until the orange-colored precipitate was obtained. Zinc oxide nanoparticles were made by adding 2 M sodium hydroxide (Sigma-Aldrich, MO, USA) to an aqueous solution of 1M zinc sulfate heptahydrate (Sigma-Aldrich, MO, USA), followed by 2 mL 0.01% polyvinyl alcohol and vigorously stirred for about 18 h. Curcumin and ZnO nanoparticles were fixed for TEM investigation by placing a drop of the suspension on carbon-coated copper grids. The samples were dried under an infrared lamp and then the images were recorded using TEM (JEOL JEM-1400Flash instrument, Tokyo, Japan). The average particle size and polydispersity index (PDI) were evaluated by Zetasizer (Malvern Instruments, Malvern, WR14 1XZ, United Kingdom).

### 2.2. Experimental Animals

The animal study was performed in the laboratory animal house of the Medical Research Institute, Alexandria University, and compliance with the policy of animal care. Eighty male Wistar rats (Medical Research Institute, Alexandria University, Alexandria, Egypt), approximately 8–10 months of age, and weighing 120–150 g, were used in this study and were kept (five rats/cage) in polycarbonate cages with stainless-steel wire led with ad libitum feeding, a constant ambient temperature of 22 ± 2 °C, a humidity of 55 ± 5%, and a light-dark cycle of 12 h. Animals were humanely cared for during all experiments, and all efforts were made to minimize animal suffering.

### 2.3. Type 2 Diabetes and Neurodegeneration Rat Model

After two weeks of acclimatization, the rats were further divided into nine groups as follows: control group, received saline; HFD/STZ-induced group, induced to establish a simultaneous T2DM and neurodegeneration disease model; HFD/STZ-Cur, model rats treated with pure curcumin at a dose of 50 mg/kg body weight (b.w.); HFD/STZ-CurNP-10, model rats treated with CurNP at a dose of 10 mg/kg b.w.; HFD/STZ-CurNP-50, model rats treated with CurNP at a dose of 50 mg/kg b.w.; HFD/STZ-Zinc sulfate, model rats treated with zinc sulfate at a dose of 50 mg/kg b.w.; HFD/STZ-ZnONP-10, model rats treated with ZnONP at a dose of 10 mg/kg b.w; HFD/STZ-ZnONP-50, model rats treated with ZnONP at a dose of 50 mg/kg b.w.; and HFD/STZ-Metformin, model rats treated with metformin at a dose of 100 mg/kg b.w. All current treatment options were orally administered to rats daily for six weeks. The doses were administered daily at the same time and prepared by dissolving suitable doses of each therapy in saline according to the body weight of each rat and then ingested by oral gavage. The induction model of type 2 diabetes and neurodegeneration was established by feeding rats HFD for eight weeks followed by intraperitoneal injection of 35 mg/kg streptozotocin dissolved in 0.1 M citrate buffer, pH 4.5. Blood samples were taken three days after injection from overnight fasting rats to measure blood glucose levels. Type 2 diabetes was defined as a fasting blood glucose level of more than 350 mg/dL.

### 2.4. Morris Water Maze Test (MWM)

The spatial learning and memory function of rats was assessed with an MWM test. The MWM contained a circular pool of 200 cm diameter and 50 cm deep, which filled with water 22 ± 2 °C, plus a 10 cm diameter escape platform submerged below the surface of the water by 1 cm. Firstly, rats were trained to seek the platform that was submerged in water and exit the pool. The test was carried out for five consecutive days and the rats went through three training trials each day. Each rat was given a trial for only 180 s to successfully find the hidden platform. The time required for each rat to detect the platform, escape latency, the distance swum, and the escape distance were measured. The shorter the escape latency, the stronger spatial learning ability. On the sixth day, for the second phase of testing, a probe test to detect memory maintenance or deficit after removal of the platform, each trial was 60 s in duration. The swimming path length, speed, and time spent in the target quadrant were measured.

### 2.5. Sample Preparation and Biochemical Evaluations

After six weeks of treatment, rats were fasted overnight, then anesthetized using sodium pentobarbital (100 mg/kg, i.p.) and blood was drawn from the abdominal aorta with a syringe puncture, and serum was separated. Some brains were immediately removed and stored at −80 °C for RNA isolation and western blot testing and others were handled by isolating the hippocampus on ice. The separated hippocampus tissue was homogenized in lysis buffer with protease inhibitor (150 mM NaCl, 1% Triton X-100, 10 mM Tris, pH 7.4) and centrifuged at 10,000× *g* at 4 °C for 15 min. The total protein contents were measured with the Lowry method [34]. The remaining part of the hippocampus was stored in 10% formalin for histopathological study. The blood glucose levels were monitored with an ACCU-CHEK active blood glucose meter. Serum insulin was performed according to the ELISA technique using a commercial kit specific for rats (MyBioSource, MBS724709) according to the manufacturer’s instructions provided with the kit. Insulin resistance estimation was performed using the homeostasis model assessment method; HOMA-IR was calculated by the following formula: plasma glucose (mg/dL) × fasting plasma insulin (IU mg/L in the fasting state divided by 405) [35]. In addition, AGEs levels were determined in serum by a specific ELISA kit (MyBioSource, MBS261131), and the assays were performed following the manufacturer’s manual.

### 2.6. Measurement of Hippocampal Oxidant and Antioxidant Biomarkers

Malondialdehyde levels were assessed in the form of thiobarbituric acid-reactive substances (TBARS) using a spectrophotometric colorimetric assay at 532 nm; the concentration of TBARS in the hippocampus was expressed as nmol/mg protein [36]. Nitric oxide (NO) was assayed according to the standard procedure of the Griess reaction, and the absorbance was read at 540 nm against the blank, then expressed as μM/mg protein [37]. GSH was examined and the generated yellow color was detected at 412 nm immediately, and the GSH concentration was reported as mg/mg sample protein [38]. Glutathione-S-Transferase (GST) activity was measured in the hippocampus at 310 nm and expressed as μmol/min/mg protein [39]. In the assay of superoxide dismutase (SOD), the absorbance was measured at 420 nm and the sample enzyme activity was reported in U/mg protein [40]. The activity of catalase (CAT) was measured by measuring the breakdown of its substrate H_2_O_2_ at 240 nm and calculating the change in absorbance per minute [41]. The activity of glutathione peroxidase (GPx) was measured and estimated using the method described previously and expressed as mol/min/mg protein [42].

### 2.7. Quantification of Serum Adipokines Levels and Hippocampal IL-6 and TNF-α

Leptin (MyBioSource, MBS012834) and adiponectin (MyBioSource, MBS068220) levels were detected in serum, while IL-6 (MyBioSource, MBS355410) and TNF-α (MyBioSource, MBS2507393) were detected in the hippocampus using rat specific ELISA kits following the manufacture’s instruction.

### 2.8. Quantification of Acetylcholine Esterase Activity (AChE), IDE, and Aβ-42 Concentrations

As previously mentioned, the activity of AChE was evaluated using Ellman’s reagent colorimetric assay [43]. AChE activity was measured at 412 nm using 0.75 mM acetylthiocholine and 0.5 mM 5,5-dithiobis (2-nitrobenzoic acid) (DTNB) in 5 mM HEPES buffer (pH 7.5). Additionally, Aβ-42 (MyBioSource, MBS726579) and insulin-degrading enzyme (IDE) (MyBioSource, MBS9139258) levels were detected in the hippocampus using rat specific ELISA kits following the manufacture’s protocol.

### 2.9. Real-Time Polymerase Chain Reaction (RT-PCR)

Total RNA was isolated from the frozen hippocampus and cortex of rats using QIAzol Lysis Reagent (Qiagen, 79306) according to the manufacturer’s instructions. Then, 1 µg of total RNA was subjected to reverse transcription in triplicate for cDNA preparation, and the PCR reaction was carried out using SYBER green master mix (Qiagen, 204143); the mRNA expression levels were calculated relative to β-actin gene’s mRNA levels using 2^−ΔΔCt^ method [44]. PCR conditions were set with an initial incubation at 95 °C for 10 min, and 40 cycles at 95 °C for 15 s, 60 °C for 1 min, and 72 °C for 40 s. Primer sequence (Sigma-Aldrich): APP (NM_019288.2), F-5′-AGAGGTCTACCCTGAACTGC-3′, R-5′-ATCGCTTACAAACTCACCAAC-3′; BACE1 (NM_019204.2), F-5′-CGGGAGTGGTATTATGAAGTG-3′, R-5′-AGGATGGTGATGCGGAAG-3′; BDNF (NM_012513.4), F-5′-ATGGGACTCTGGAGAGCGTGAA-3′, R-5′-CGC CAGCCA ATTCTC TTT TTGC-3′; ADAM10 (NM_019254.1), F-5′-GTTAATTCTGCTCCTCTCCTGG-3′, R-5′-TGGATATCTGGGCAATCACAGC-3′; Bcl-2 (NM_016993.2), sense: 5′-GCAGCTTCTTTCCCCGGAAGGA-3′, antisense: 5′-AGGTGCAGCTGACTGGACATCT-3′; Bax (NM_017059.2), sense: 5′-AACTTCAACTGGGGCCGCGTGGTT-3′, antisense: 5′-CATCTTCTTCCAGATGGTGAGCGAG-3′; β-actin (NM_031144.3), sense: 5′-TGAGAGGGAAATCGTGCGT-3′, anti-sense 5′-TCATGGATGCCACAGGATTCC-3′.

### 2.10. Western Blotting

The frozen hippocampus tissue was homogenized in lysis buffer (100 mM NaCl, 100 mM EDTA, 0.5% Nonidet p-40, 0.5% Na-deoxycholate, 10 mM Tris, pH 7.5, containing protease inhibitors). After centrifuging the sample for 10 min at 2000× *g* and 4 °C, the supernatant was collected, and the protein content was determined. The protocol for Western blot testing was followed as stated previously [45]. Briefly, 50 µg of protein samples were mixed with 2X loading buffer and boiled at 100 °C for 10 min. The proteins were then separated using SDS-PAGE (12%) and transferred to nitrocellulose membranes. Then the membranes were blocked by soaking for 1 h at room temperature in a solution of 5% nonfat milk in TBST buffer (10 mM Tris–HCl, pH 8.0, 150 mM NaCl, and 0.2 percent Tween-20). The membranes were then treated with primary antibodies overnight at 4 °C (β-actin (NB600-501), p-p38 MAPK (9215), p38 MAPK (9211), p-Erk1/2 (4377), ERK1/2 (9102), MEK1/2 (9122), p-MEK1/2 (Ser 217/221; 9154)), then washed with TBST after primary incubation and then incubated with secondary antibodies for 1 h at room temperature. The signal was developed using a TMB Western Blotting Detection kit. Image Lab 6.1 software (BioRad) was then used to quantify the produced bands.

### 2.11. Histological Analysis

The fixed brain tissues in 10% formalin were processed for dehydration in ascending grades of alcohol, followed by impregnation. The specimens were then embedded in paraffin and allowed to solidify at room temperature. Using a rotatory microtome, serial sections of 5 μm thick were cut. After that, sections were stained with Haematoxylin and Eosin (H&E) and observed for histopathological changes.

### 2.12. Statistical Analysis

Results are shown as mean ± SEM (standard error of the mean) for eight rats in each group. Significant differences between groups were compared by one-way analysis of variance (ANOVA), followed by Tukey’s post hoc test for multiple comparisons; with a *p*-value < 0.01, the difference between the groups was considered statistically significant.

## 3. Results

### 3.1. Characterization of ZnONP and CurNP

TEM analysis displayed that ZnONP had a diameter between 11 and 22 nm while CurNP had a diameter between 44 and 52 nm and both nanoparticles had a spherical shape (Figure 1A). Additionally, the particle size of ZnONPs and CurNPs (Figure 1B) was 21.00 nm and 58.80 nm, respectively, confirming the results obtained from TEM analysis. Where the PDI was (0.201 ± 0.01) for ZnONP and (0.141 ± 0.01) for CurNP. CurNP and curcumin FTIR spectra were measured in the region of 4000–400 cm^−1^. The stretching vibration of hydrogen-bonded OH found in curcumin correlates to the intense band at wavenumber 3508 cm^−1^. The frequency of aromatic CH was determined to be 1628 cm^−1^. The aromatic stretching vibrations of the benzene ring are represented by 1427 cm^−1^. The intense characteristic band centered at 1507 cm^−1^ represents the stretching vibration of the conjugated carbonyl (C=O). The stretching vibration of hydrogen-bonded OH observed in nano curcumin corresponds to the broad, strong band at wavenumbers 3268 and 3268 cm^−1^. The stretching vibrations of the benzene ring are shown by bands at 1455 and 1430 cm^−1^. The characteristic band centered at 1513 cm^−1^ represents the stretching vibration of the conjugated carbonyl (C=O). Stretching vibrations of C=C bonds are shown at 1598 (Figure 2C). The chain absorption peaks in the FTIR spectra of ZnONP were 400 and 4000 cm^−1^. The C=C stretch of alkenes or the C=O stretch of amides is represented by the absorption peaks at 1631, 1505, and 1381 cm^−1^. Stretching vibrations of the C–O bond are responsible for the bands at 1076 cm^−1^. Peaks at 692 and 899 cm^−1^ may also be related to C–N stretching amine groups. Furthermore, in the absorption peaks of 3433 and 3396 cm^−1^, hydroxyl group stretching can be noticed. The primary absorption bands 422, 434, and 470 were assigned to the Zn–O bond and are distinctive of it (Figure 2D).

### 3.2. Effects of CurNP and ZnONP on Memory Deficits in T2DM-Induced Rat Models

The MWM test was used to determine the time it took rats to identify and mount a water maze platform (escape latency) and the total swimming distance they covered before escaping (escape distance). There was no significant difference in escape latency (Figure 2A, *p* < 0.01) or distance (Figure 2B, *p* < 0.01) for either group for the first three days. On the fourth day, however, the model group had significantly longer escape latency and distance than the controls (*p* < 0.01). Furthermore, compared to model rats, the rats receiving current medications had considerably lower escape latency and distance (*p* < 0.01). However, rats given 50 mg/kg ZnONP or 10 mg/kg CurNP had significantly longer escape latency and distance than rats given other treatments (*p* < 0.01). Probe testing showed that the model group had a considerably shorter swim time (Figure 2C) and distance swum (Figure 2D) in the target quadrant (represented as a percentage of total time and distance swum) than the control group (*p* < 0.01). Treatment groups had considerably higher percentages of time and distance swum compared to model rats (*p* < 0.01). However, rats given 50 mg/kg ZnONP or 10 mg/kg CurNP spent significantly less time and swam significantly lesser distances than rats given other treatments (*p* < 0.01).

### 3.3. Blood Glucose, Insulin, and AGEs Levels

The fasting glucose and insulin levels in type 2 diabetic rats after HFD/STZ induction displayed a significant increase versus the control group (*p* < 0.01). In contrast, after the treatment period with all current treatment options, the fasting glucose and insulin levels were reduced significantly relative to the untreated group (*p* < 0.01). Changes examined in levels of glucose and insulin are also represented in the HOMA-IR index; marked differences between model and healthy groups were found in HOMA-IR, and the treatment of model rats for six constitutive weeks was associated with a significant decrease in HOMA-IR index. In addition to this, T2DM may promote AD pathology through chronic hyperglycemia, which is associated with elevation of AGEs levels in serum of T2DM-induced rats compared to healthy control (*p* < 0.01). All treatments of T2DM may affect AD indirectly via effects on circulating levels of glucose and insulin. In our study, the administration of curcumin, zinc sulfate, two doses of CurNP and ZnONP, as well as metformin, resulted in a marked decrease in AGEs levels relative to untreated rats; the best results were shown in groups treated with 10 mg/kg CurNP and 50 mg/kg ZnONP, which reached levels comparable to those of the control group (Table 1, *p* < 0.01).

### 3.4. Hippocampal Concentrations of Oxidant and Antioxidant Biomarkers

As shown in Table 2, there was a marked elevation in hippocampal levels of NO and TBARS relative to controls (*p* < 0.01). Additionally, simultaneous marked decreases in GST, CAT, GPx, and SOD activities, as well as GSH content, were noticed in rats from the type 2 diabetes model relative to controls (*p* < 0.01). In contrast, a marked decrease in hippocampal TBARS and NO levels and concurrent increases in GST, CAT, GPx, and SOD activities, as well as GSH content, were observed in all treated rats, especially 10 mg/kg CurNP and 50 mg/kg ZnONP, relative to model groups (*p* < 0.01). These results suggest a better aptitude of CurNP at a dose of 10 mg/kg to fight oxidative stress in the brain of rats.

### 3.5. Alteration of Concentrations of Serum Adipokines and Hippocampal Inflammatory Mediators

These results indicate a significant decrease in the serum adiponectin levels of type 2 diabetes-induced rats versus the control group (*p* < 0.01), while serum leptin levels were found to be increased in type 2 diabetes-induced rats as compared to control animals (*p* < 0.01). Treatment of diabetic rats with ZnONP and CurNP improved serum adiponectin levels significantly in type 2 diabetes-induced rats versus untreated rats (*p* < 0.01). In contrast, administration of all current treatment options to type 2 diabetes-induced rats produced a significant reduction in serum leptin levels compared to the untreated group (*p* < 0.01), and 10 mg/kg CurNP and 50 mg/kg ZnONP showed a more efficient decrease in serum leptin levels than other doses (Table 3, *p* < 0.01). Further, to investigate the effect of CurNP and ZnONP on proinflammatory cytokines, IL-6 and TNF-α protein levels were detected in all groups. Our results show an increase in TNF-α and IL-6 protein levels in the hippocampus of model rats compared with the healthy control rats, TNF-α and IL-6 levels in type 2 diabetes-induced rats increased 10.2-fold and 3.9-fold in the hippocampus, respectively (*p* < 0.01), while among all current treatment options, 10 mg/kg CurNP and 50 mg/kg ZnONP treatment significantly attenuated the increased TNF-α and IL-1β levels in type 2 diabetes-induced rats relative to the untreated model (Table 3; *p* < 0.01).

### 3.6. Regulation of AChE Activity, IDE Level, and Aβ-42 Clearance in the Hippocampus of Rats

As presented in Table 4, AChE activity in the hippocampus was significantly increased in the type 2 diabetes model group versus healthy controls (*p* < 0.01). Alternatively, a significant decrease in AChE activity was found in the hippocampus of rats obtaining CurNP and ZnONP treatment versus model rats (*p* < 0.01). Consistently, IDE levels tended to be lower in the hippocampus of rats with type 2 diabetes-induced neurodegeneration compared to control rats (*p* < 0.01). The treatment of the induced groups with ZnONP and CurNP resulted in an increase in hippocampal IDE levels that was similar to that for the control group compared to untreated rats (Table 4, *p* < 0.01). To date, Aβ-42 production and aggregation are considered the center of attention of neurodegeneration models like AD. In our study, a significant elevation of Aβ-42 levels in the hippocampus of rats with type 2 diabetes-induced neurodegeneration was detected compared to healthy rats (*p* > 0.01). The treatment of the induced groups with ZnONP and CurNP, particularly those with 10 mg/kg CurNP and 50 mg/Kg ZnONP, exhibited a marked decrease in hippocampal Aβ-42 levels with values near to the control group.

### 3.7. Gene Expression Profile of APP, BACE-1, BDNF, and ADAM-10 in the Hippocampus of Rats

Type 2 diabetes-induced neurodegeneration in rats revealed a significant upregulation of gene expression of APP and BACE-1 when compared with control rats (*p* < 0.01). Interestingly, these elevations were markedly suppressed following the administration of curcumin, zinc sulfate, two doses of CurNP, ZnONP, and metformin to model rats versus healthy rats (*p* < 0.01). Conversely, gene expression of BDNF and ADAM-10 were significantly downregulated in the hippocampus of model rats when compared with the control group (*p* < 0.01). Further, administration of curcumin, zinc sulfate, two doses of CurNP, ZnONP, and metformin displayed a marked upregulation of gene expression of BDNF and ADAM-10 compared to model rats (Figure 3, *p* < 0.01). Of all treatment options, 10 mg/kg CurNP and 50 mg/kg ZnONP significantly ameliorated the gene expression of the evaluated genes (Figure 3).

### 3.8. Effect of CurNP and ZnONP on an Apoptotic Pathway in the Hippocampus and Cortex of Rats

Under stress conditions, Bcl-2 expression levels in the hippocampi and cortices of rats with type 2 diabetes-induced neurodegeneration were significantly low (*p* < 0.01), while Bax expression was significantly high compared with the control group (*p* < 0.01). At the end of the CurNP and ZnONP treatments of model rats, the proapoptotic member, Bax expression was significantly suppressed along with increased Bcl-2 expression in both hippocampi and cortices (Figure 4A,B, *p* < 0.01).

### 3.9. p38-MAPK Signaling Cascade and Tau Protein in the Hippocampus of Rats

It has been reported that the MAPK pathway, p38, ERK, and MEK activate many signaling cascades including inflammation and apoptosis. Consequently, we here examined the protein phosphorylation levels of MAPK signaling in the hippocampus of rats by western blotting. As shown in Figure 5, the levels of phosphorylated-p38MAPK, phosphorylated-ERK 1,2 (p-ERK), and phosphorylated-MEK (p-MEK) in rats with type 2 diabetes-induced neurodegeneration were increased significantly compared with the control group (*p* < 0.01). After CurNP and ZnONP treatment, the phosphorylation levels of p-p38MAPK, p-ERK, and p-MEK were markedly decreased compared with those of the untreated group (*p* < 0.01). Our data indicated that 10 mg/kg CurNP or 50 mg/kg ZnONP might be associated with the regulation of neuroinflammation and apoptosis in the hippocampus of type 2 diabetes-induced rats at least partly, by adjusting the activation of the MAPK signaling pathway. Additionally, type 2 diabetes-induced rats revealed significant hyperphosphorylation of tau proteins when compared with the control rat group (*p* < 0.01). Interestingly, this elevation was markedly suppressed following administration of all current treatment options to model rats, with marked inhibition of tau phosphorylation observed in groups treated with 10 mg/kg CurNP or 50 mg/kg ZnONP (Figure 5A–E).

### 3.10. Histopathological Study

The histopathological study of the hippocampi was seen in Figure 6. Paraffin section photomicrographs of control brains show the hippocampus area with arranged pyramidal cells with large pyramidal nuclei and minimum vacuolated cytoplasm, few pyknotic neuropil cells, and homogenous brain tissue (Figure 6A). Conversely, the H&E-stained sections of type 2 diabetes-induced group display a marked proliferating pyramidal cell forming gland feature and have dark nuclei with minimum vacuolated cytoplasm and many pyknotic neuropil cells, marked congested, dilated blood vessels and capillaries, beside edema and reduction of the myelinated sheath, indicating AD induction (Figure 6B). Moreover, paraffin section photomicrographs of rat brains treated with curcumin-50 show the hippocampus area with some pyramidal cells, hyperchromatic nuclei, and marked vacuolated cytoplasm with many pyknotic pyramidal cell and neuropil cells, as well as reduction of the myelinated sheath, and a mildly dilated and hemorrhaged blood capillary was seen (Figure 6C). Besides, the brain photomicrographs of the induced group treated with CurNP-10 shows the hippocampus region with dark pyramidal nuclei and low vacuolated cytoplasm (normal architecture) and few pyknotic pyramidal cells; few neuroglial cells have rounded nuclei; prominent nucleolus, marked hemorrhage dilated blood vessels, and capillaries with the homogenous field of myelinated sheath and edema were seen (Figure 6D). In addition, the brain photomicrographs of the induced group treated with CurNP-50 show hippocampi with few neuroglial cells and proliferating pyramidal cells, having dark pyramidal nuclei and mild vacuolated cytoplasm. Furthermore, rearrangement of a pyramidal cell for the organized hippocampus region and a few necrotic ones were noticed. A mildly dilated and hemorrhaged blood vessel with homogenous brain tissue was seen (Figure 6E). In addition, rat brains treated with zinc sulfate-50 showed a hippocampus area with some pyramidal cells, hyperchromatic nuclei with homogenous cytoplasm, few necrotic and many pyknotic pyramidals, as well as neuroglial cells. A mildly dilated and hemorrhaged blood capillary with a mildly degenerative myelinated sheath was also seen (Figure 6F). Furthermore, the brain photomicrographs of the ZnONP-10-treated group showed a hippocampus with regenerative pyramidal cells and hyperchromatic nuclei, with mild vacuolated cytoplasm and few necrotic neuroglial cells. A mildly dilated blood capillary with an area of homogenous brain tissue was also seen (Figure 6G). In addition to this, the brain photomicrographs for the ZnONP-50-treated group showed a hippocampus with the normal features of many neuropil cells and few pyknotic ones, and mild proliferating pyramidal cells with hyperchromatic nuclei and mildly vacuolated cytoplasm. A mildly dilated and hemorrhaged blood vessel with homogenous brain tissue was also observed (Figure 6H). The brain photomicrographs of metformin-treated rats showed the hippocampus region with many neuroglial cells with rounded nuclei and a prominent nucleolus, as well as necrotic neuroglial cells. Many necrotic pyramidal cells and others with small nuclei (atrophied pyramidal cells) and pyknotic pyramidal cells were seen. Markedly dilated and hemorrhaged blood vessels and capillaries were also noticed, as was the reduction of a myelinated sheath in the necrotic area (Figure 6I).

## 4. Discussion

Accumulating data have established that T2DM-induced cognitive deficits increase the risk for AD. Further, T2DM-associated complications, such as oxidative stress and inflammation, are thought to play a critical role in cognitive decline. Therefore, the current study was aimed to evaluate novel therapeutic effects and the mechanisms of nanoparticles from a natural source like CurNP, as well as metal oxide-base nanomaterials like ZnONP, on the hippocampus of T2DM-induced rats.

T2DM develops in case of insulin resistance and dysfunction of insulin secretion [46]. We and others have shown that the HFD/STZ model appears to produce a degree of neurodegeneration and mimics pre-AD symptoms in the hippocampus, including memory loss, insulin resistance, Aβ aggregation, and tau hyperphosphorylation. It is also considered a useful experimental model to study the activity of hypoglycemic agents [47,48].

Constant hyperglycemia, hyperinsulinemia, and a significant elevation in HOMA-IR were observed in model rats and are consistent with our previous results [47]. The long-term administration of either CurNP and ZnONP is effective in reducing T2DM-associated alterations through decreasing blood glucose and insulin levels and enhancing HOMA-IR. These results are not surprising considering curcumin’s hypoglycemic effect on pancreatic cells and insulin sensitivity, which have been extensively studied [49]. Furthermore, ZnONP can improve serum insulin levels, glucose utilization, and metabolism by influencing hepatic glycogenesis and possibly acting on the insulin signaling pathway [50,51,52]. Nevertheless, a preceding study presented an adverse effect of short-term administration of ZnONP (100 mg/kg) that increased blood glucose levels in diabetic and healthy rats depending on the dose and route of administration [53].

Our results revealed that hyperglycemia leads to tissue damage by several mechanisms, including the AGEs formation. Furthermore, accumulating evidence suggested that AGEs play a key role in the pathogenesis of neurodegenerative disorders, such as AD and diabetic neuropathy, a diabetes-related complication [54]. Until now, it has been well recognized that blood proteins, including IgG, are glycated in AD patients by a variety of reducing sugars and metabolites, manifesting in AGEs formation leading to protein structure and function distortion [55]. The current findings show an increase in AGEs levels in serum of model rats which confirm our previous findings [47]. All current therapies, especially CurNP and ZnONP, according to our findings, reduced the production of AGEs. Therefore, both nanoparticles may have therapeutic potential in delaying disease development and/or reversing disease pathogenesis in AGEs-related neurodegenerative disorders.

Interestingly, the hippocampus is an important structure in rats and humans, and it plays a crucial integrative role in learning and memory [56]. As a result, hyperglycemia-induced alterations in the structure of the hippocampus may represent the pathological basis of cognitive deterioration in diabetic rats [57]. Our behavioral analyses demonstrate that HFD/STZ induction prolonged escape latency and distance in MWM training, indicating spatial learning deficiencies. CurNP and ZnONP treatments reduced the escape latency and distance of model rats, protecting them against HFD/STZ-induced spatial learning impairments. In the probe trial, shorter times spent in the target quadrant and fewer platform zone crossings were seen in model rats, indicating impairments in spatial memory. CurNP and ZnONP treatments protected the spatial memory of model rats; they spent more time in the target quadrant and crossed the platform zone twice more. Our study suggests that CurNP and ZnONP may protect the spatial and memory functions in the T2DM rat model.

Certainly, the combination of the pressure from oxidative stress with lowered antioxidant defense creates a harmful effect that contributes to disrupting the functions of cells and leads to cell death. In the current work, the antioxidant capacity parameters (SOD, GPx, CAT, and GST enzyme activities, as well as GSH level) and typical oxidative stress biomarkers (TBARS and NO levels) were used to investigate the redox profile in the hippocampus of the rats model. The results demonstrated a remarkable elevation in oxidative stress biomarkers, with a significant reduction of enzymatic and non-enzymatic antioxidants in the hippocampus of model rats. These findings are consistent with prior research that linked amyloid plaque formation to oxidative stress, including lipid peroxidation markers as TBARS [58]. Additionally, hyperglycemia activates multiple signaling pathways, which lead to increased ROS production and induce insulin resistance [59]. Indeed, oxidative stress is one of the initial signs of AD disease and is the first consequence of Aβ accumulation in the brain [60]. Treatment of model rats with 10 mg/kg CurNP exhibited the most potent effect in terms of amelioration of hippocampal redox profiles, followed by 50 mg/kg ZnONP, compared with other doses of nanoparticles and classical curcumin and zinc sulfate. Curcumin and CurNP are considered antioxidant agents and could enhance the expression of antioxidant enzymes and thus increase total antioxidant capacity [61]. Moreover, our data for diabetic rats demonstrated that treatment with ZnONP efficiently restored GPx, SOD, and CAT activities by increasing the biosynthesis of GSH or reducing oxidative stress [62,63,64]. Conversely, as mentioned before, the intraperitoneal injection of 25 mg/kg ZnONP had no significant effect on the levels of CAT, SOD, and GSH in the brain, while treatment with ZnONP at a high dose decreased brain GSH and SOD and increased brain MDA levels [65].

Oxidative stress excites the translocation of the transcription factor nuclear factor kappa B (NF-κB) to the nucleus, which adjusts pro-inflammatory gene expressions, such as TNF-α and IL-6 [66]. By further analyzing hippocampal inflammatory cytokines, we were able to pin down the anti-inflammatory effect of nanoparticles. We showed that the administration of our current treatments produced a significant reduction in TNF-α and IL-6 levels, with the maximum inhibition level in both 10 mg/kg CurNP and 50 mg/kg ZnONP treated groups. Curcumin and CurNP confirmed a reduced level of inflammatory cytokines: TNF-α and IL-6 [67]. Curcumin prohibited the elevation of the cytokines since it can decrease the inflammatory reactions by interfering with NF-κB activation [68]. Likewise, it has been found that zinc and ZnONP were attributed to the anti-inflammatory effect of this ion [62,69], which inhibits the gene expression of inflammatory cytokines, including TNF-α and IL-6, which are known to produce ROS [70,71]. In contrast to our study, it was demonstrated that daily administration of 100 mg/kg ZnONP significantly increased the concentrations of TNF-α and IL-1β [72].

Our findings showed high leptin levels and significant adiponectin reduction in model rats, and after six weeks of supplementation of CurNP, ZnONP, a significant amelioration of serum adipokines was observed. Hence, the protective curcumin effects associated with adiponectin expression upregulation were suggested [61], these inhibiting lipogenic expression and adipose tissue inflammation [73]. Inflammation due to visceral adiposity is improved and this effect comes with an increase in adiponectin and an improvement in sensitivity to insulin [74]. The ZnONP-enhanced mechanism is remarkable, and this huge effect is achieved through a direct entry into the intracellular region, which leads to the adjustment of many enzymes, adipogenesis, and transcription factors.

Further, the anti-apoptotic Bcl-2 and pro-apoptotic Bax are two critical molecules involved in cell death [75]. The enhanced expression of Bax and decreased expression of Bcl-2 following the HFD/STZ induction observed in the current study suggest that neuro-apoptosis is elevated by T2DM induction in the hippocampi and cortices of induced rats. The equilibrium of the antiapoptotic (represented by Bcl-2) and pro-apoptotic (represented by Bax) molecules were significantly changed post-treatment with CurNP and ZnONP, which effectively inhibited apoptosis in the hippocampus and cortex and increased neuronal survival. Further studies of the apoptotic pathways like caspase(s) and cytochrome c are needed to determine their involvement in our pathway, like the mitochondrial-dependent intrinsic apoptotic pathway, as reported in other studies [76].

The regulation of cholinergic function is a critical aspect of the management of AD development. We supported this by showing a partial inhibition in AChE activity in the model rat after administration of both nanoparticles [77,78]. It is noteworthy that the molecular pathogenesis of AD is underlain by accelerated Aβ production, aggregation, and compromised Aβ clearance [79]. Current results revealed an elevation in the Aβ-generating enzyme BACE1, which is responsible for Aβ production [80], and depletion in Aβ-degrading enzyme (IDE), which is responsible for Aβ clearance [81]. As stated previously, the level of Aβ-42 in the hippocampus of HFD/STZ-induced rats showed a significant elevation compared to control rats, proving the inability of the brain to clear the Aβ deposits in T2DM. This may be due to hyperinsulinemia perceived in diabetic rats and a struggle between degradation of Aβ-42 and insulin for the same substrate, IDE [81].

As it happens, the administration of CurNP and ZnONP significantly downregulated the expression of APP and BACE-1. In addition to this, they decrease the Aβ-42 levels and significantly elevate the IDE concentration in the hippocampus of rats, with the maximum effective results, compared with curcumin, zinc sulfate, and metformin. Numerous earlier studies have shown that natural antioxidants, such as CurNP, prevent neuronal death occurring in AD, and lead to a reduction in Aβ plaque deposition [24]. These results were linked with a marked upregulation of gene expression of BDNF and ADAM-10, providing a new strategy for attenuating neurodegeneration developed in T2DM [82].

The present study demonstrates that the HFD/STZ rat model is a well-established model for evaluating the efficacies of medication intended for the treatment of AD, which causes cognitive deficits, microglial activation, and phosphorylation of hippocampal MAPK/ERK. The expression of MAPK family proteins (MEK, ERK1/2, and p38-MAPK) was studied to further understand the molecular pathways related to neuroprotection by naturally occurring nanoparticles, CurNP, and the biomedical metal oxide nanoparticle, ZnONP. The levels of phosphorylated p38-MAPK, ERK1/2, and MEK in the hippocampus of rats following HFD/STZ induction were considerably increased [83]. MAPK signaling is important for synaptic plasticity and hippocampus-dependent memory [84]; also, p38-MAPK is an important kinase in tau phosphorylation [85]. In the hippocampus and microglia of rats under pre-AD, hyperphosphorylation of the MAPK pathway, besides tau-hyperphosphorylation, has been seen [86]. Our results in the present study showed that our current treatments inhibit microglial activation by decreasing the phosphorylation of the MAPK family (p38-MAPK, ERK1/2, MEK) and phosphorylation of tau protein in the hippocampus of HFD/STZ-induced rats [87]. Consistent with our results, as reported before, CurNP plays a role in inhibiting the phosphorylation of MAPKs-regulated IL-6 and TNF-α production and suppresses the phosphorylation of tau protein [88]. In addition to this, the results showed a significant effect of ZnONP at a dose of 50 mg/kg on the MAPK/ERK pathway via the marked inhibition of its phosphorylation, and this is the first time the effect of ZnONP on the MAPK pathway in the hippocampus of T2DM-induced rats has been studied. Overall, our results indicated that the MAPK signaling pathway appears to be implicated in CurNP and ZnONP-mediated neuroprotection.

The damage in any cell in the hippocampus can cause gross effects on the learning process of the individual. After histopathological examination of normal and diabetic hippocampal sections, results revealed that diabetes had a significant effect in the form of cell death in various areas, as well as disruption of normal layer organization. This was associated with clumping of neuronal processes, a sign of neuronal injury [89]. Fortunately, these changes were significantly improved by the administration of CurNP and ZnONP, showing well-preserved pyramidal cells in both groups, which confirms the neuroprotective effect of both nanoparticles on brain cells through the improvement of neurogenesis.

## 5. Conclusions

In conclusion, the findings of the present study demonstrate that naturally occurring CurNP and the biomedical metal-oxide ZnONP inhibit HFD/STZ-induced neuro-apoptosis, neuroinflammation, cognitive dysfunction, amyloidogenesis, and tau hyperphosphorylation in the hippocampi of rats. These exhibited modifications likely occur by modifications to the critical pathways involved in these complications, such as p38-MAPK/ERK. In general, new information has been discovered that adds to the puzzle of nanoparticle use in biomedicine. As a result of our findings, taking 10 mg/kg naturally CurNP or 50 mg/kg metal oxide ZnONP may potentially reduce diabetic complications-induced neurotoxicity and aid in proper care, particularly in the case of T2DM. However, more research is needed to confirm this, particularly regarding the use of ZnONP.

## Figures and Tables

**Figure 1 pharmaceutics-13-01937-f001:**
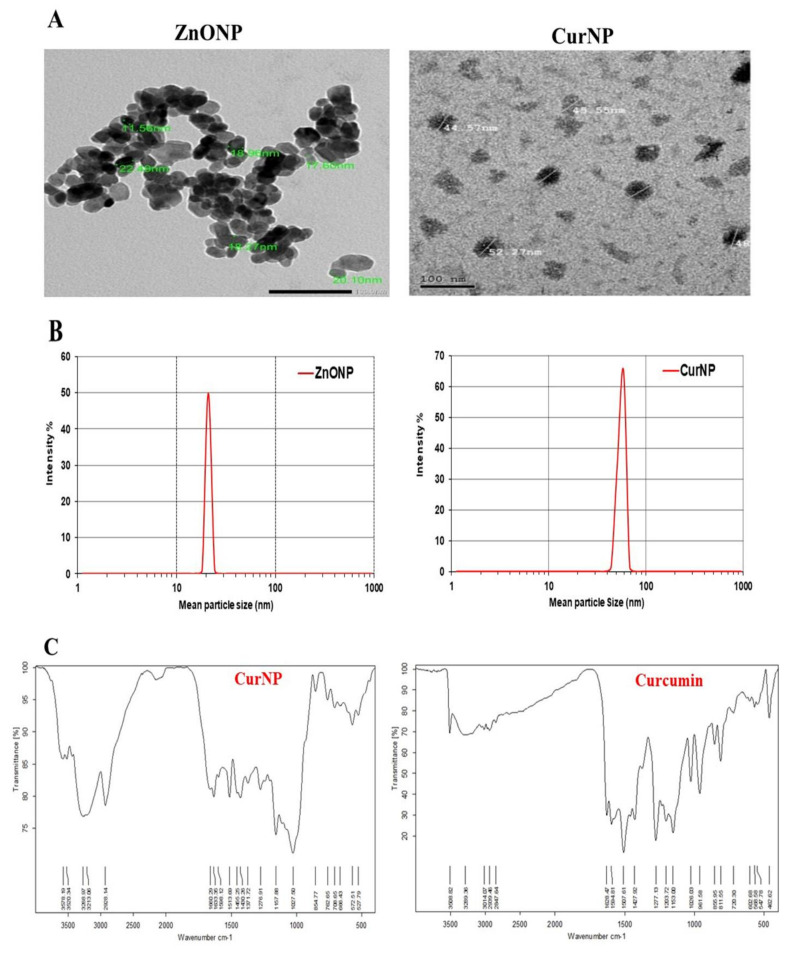
Characterization of ZnONP and CurNP. (**A**) TEM analysis of nanoparticles. (**B**) The particle size of nanoparticles. (**C**) FTIR analysis of curcumin and CurNP. (**D**) FTIR analysis of ZnONP.

**Figure 2 pharmaceutics-13-01937-f002:**
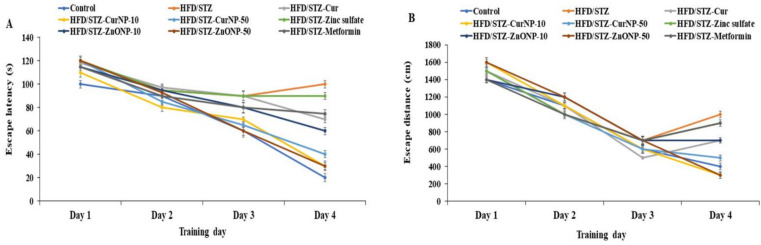
Effects of ZnONP and CurNP on memory impairment in rats. The Morris water maze (MWM) was used to test rats’ memories by measuring (**A**) escape latency and (**B**) escape distances. A probe test was used to analyze the maintenance of memory in the MWM by measuring the percentage of (**C**) time spent and (**D**) distance swum in the target quadrant. Three independent experiments were performed. Data are expressed as mean ± SEM (*n* = 8); means with different letters (a–e) in each bar are significantly different (*p* < 0.01), in Figure 2C the largest data value take the letter (a) and the smallest data value take the letter (e) and in Figure 2D the largest data value take the letter (a) and the smallest data value take the letter (c).

**Figure 3 pharmaceutics-13-01937-f003:**
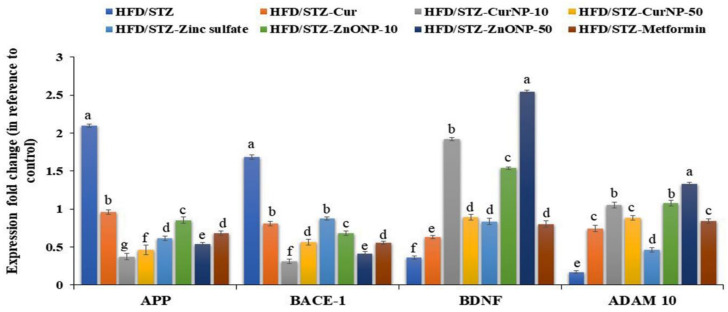
Gene expression profile of APP, BACE-1, BDNF, and ADAM-10 in the hippocampus of rats. Data are expressed as mean ± SEM (*n* = 3); means for the same parameter with different letters (a–g) in each bar are significantly different (*p* < 0.01), the largest data value takes the letter (a) and the smallest data value takes the letter (g).

**Figure 4 pharmaceutics-13-01937-f004:**
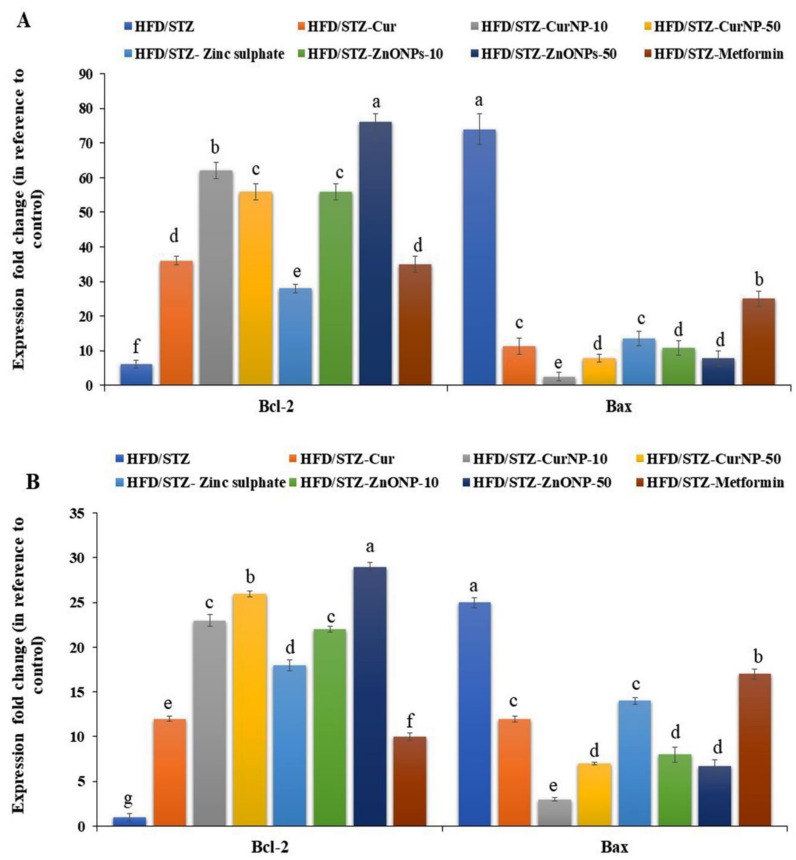
Gene expression profile of Bcl-2 and Bax in (**A**) hippocampus and (**B**) cortex of rats. Data are expressed as mean ± SEM (*n* = 3); means for the same parameter with different letters (a–g) in each bar are significantly different (*p* < 0.01), the largest data value takes the letter (a) and the smallest data value takes the letter (g).

**Figure 5 pharmaceutics-13-01937-f005:**
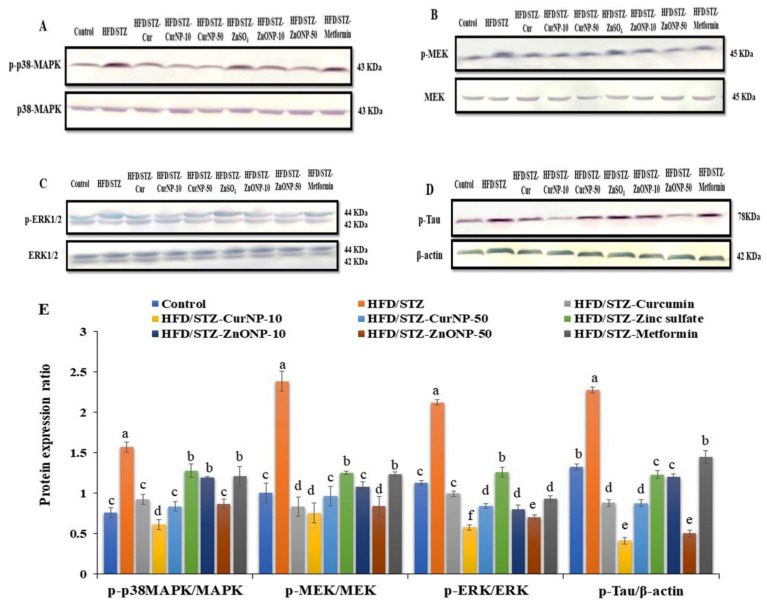
Protein expression ratio profile. (**A**) p-p38MAPK/p38MAPK, (**B**) p-ERK/ERK, (**C**) p-MEK/MEK, (**D**) p-Tau/β-actin in the hippocampus of rats. (**E**) Quantitative analysis of p-p38MAPK/p38MAPK, p-ERK/ERK, p-MEK/MEK, and p-Tau/β-actin in the hippocampus of rats. Data are expressed as mean ± SEM (*n* = 3); means for the same parameter with different letters (a–f) in each bar are significantly different (*p* < 0.01), the largest data value takes the letter (a) and the smallest data value takes the letter (f).

**Figure 6 pharmaceutics-13-01937-f006:**
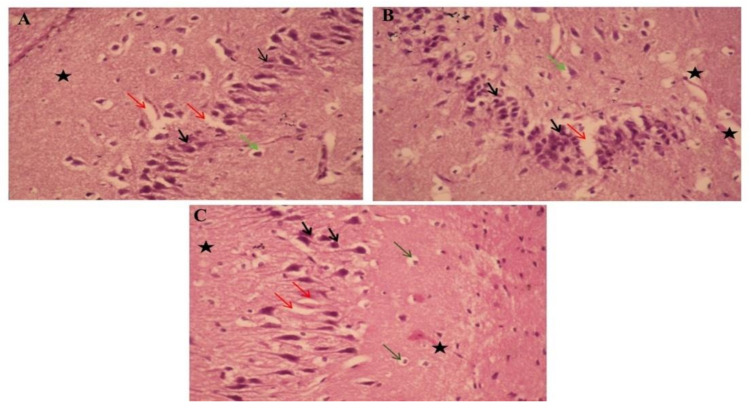
The effects of ZnONP and CurNP on the histology of hippocampi in rat groups. Hippocampi of rats were evaluated via hematoxylin-eosin (H&E) staining at 400 X magnification. (**A**) The control group: a photomicrograph of the hippocampus showing arranged pyramidal cells (black arrows) with minimum vacuolated cytoplasm (Red arrows), a few pyknotic neuropil cells (Green arrows), and homogenous brain tissue (black asterisk). (**B**) Untreated HFD/STZ-induced group: a photomicrograph of the hippocampus showing marked proliferating pyramidal cells with dark nuclei (black arrows) with minimum vacuolated cytoplasm (red arrows) and many pyknotic neuropile cells (green arrows), and reduction of the myelinated sheath (black asterisk). (**C**) HFD/STZ-induced rats treated with curcumin: a photomicrograph of the hippocampus showing some pyramidal cells with hyperchromatic nuclei (black arrows) and marked vacuolated cytoplasm (red arrows) and many neuropile cells (green arrows), with mildly dilated and hemorrhaged blood capillaries (black asterisk). (**D**) HFD/STZ-induced rats treated with CurNP-10: a photomicrograph of the hippocampus showing normal features of many neuropil cells and few pyknotic ones (green arrows), mildly proliferating pyramidal cells with hyperchromatic nuclei (black arrows) and mildly vacuolated cytoplasm (red arrows) with homogenous brain tissue (black asterisk). (**E**) HFD/STZ-induced rats treated with CurNP-50: a photomicrograph of the hippocampus showing pyramidal cells having dark pyramidal nuclei (black arrows) and mildly vacuolated cytoplasm (red arrows) with few necrotic ones (green arrows) and homogenous brain tissue (black asterisk). (**F**) HFD/STZ-induced rats treated with zinc sulfate: a photomicrograph of the hippocampus showing some pyramidal cells with hyperchromatic nuclei and homogenous cytoplasm (black arrows), many neuroglial cells (green arrows), with mildly degenerative myelinated sheaths (black asterisk). (**G**) HFD/STZ-induced rats treated with ZnONP-10: a photomicrograph of the hippocampus showing pyramidal cells, dark pyramidal nuclei (black arrows) and low vacuolated cytoplasm (normal architecture) (red arrows), prominent nucleolus (green arrows), and markedly hemorrhaged dilated blood vessels and edema (black asterisk). (**H**) HFD/STZ-induced rats treated with ZnONP-50: a photomicrograph of the hippocampus showing hyperchromatic nuclei (black arrows) with mild vacuolated cytoplasm (red arrows), and pyknotic pyramidal (green arrows), with the area of homogenous brain tissue (black asterisk). (**I**) HFD/STZ-induced rats treated with metformin: a photomicrograph of the hippocampus showing many prominent nucleoli (green arrows), pyknotic pyramidal cells (black arrows), and reduction of myelinated sheaths in the necrotic area (black asterisk).

**Table 1 pharmaceutics-13-01937-t001:** Effects of CurNP and ZnONP on the serum fasting blood glucose, insulin, HOMA-IR, and AGEs levels.

Groups	Fasting Blood Glucose (mg/dL)	Insulin (µU/mL)	HOMA-IR	Serum AGEs (ng/mg Protein)
Control	106.80 ± 0.66 ^c^	14.72 ± 0.22 ^d^	3.88 ± 0.06 ^e^	12.34 ± 0.03 ^d^
HFD/STZ	489.20 ± 1.80 ^a^	169.60 ± 1.40 ^a^	204.85 ± 1.70 ^a^	33.21 ± 0.01 ^a^
HFD/STZ-Cur	88.80 ± 0.58 ^d^	57.24 ± 0.30 ^c^	12.55 ± 0.12 ^c,d^	22.41 ± 0.01 ^b^
HFD/STZ-CurNP-10	89.20 ± 0.37 ^d^	71.24 ± 0.56 ^b^	15.68 ± 0.12 ^c^	14.31 ± 0.06 ^c,d^
HFD/STZ-CurNP-50	92.20 ± 0.86 ^d^	68.76 ± 0.40 ^b^	15.65 ± 0.13 ^c^	17.99 ± 0.02 ^c^
HFD/STZ-Zinc sulfate	119.00 ± 1.20 ^b^	70.58 ± 0.50 ^b^	20.74 ± 0.33 ^b^	25.41 ± 0.02 ^b^
HFD/STZ-ZnONP-10	75.60 ± 0.50 ^e^	60.58 ± 0.45 ^c^	11.30 ± 0.12 ^d^	20.32 ± 0.04 ^b^
HFD/STZ-ZnONP-50	89.80 ± 0.37 ^d^	69.34 ± 0.60 ^b^	15.37 ± 0.14 ^c^	15.67 ± 0.03 ^c^
HFD/STZ-Metformin	70.60 ± 0.93 ^e^	61.22 ± 0.26 ^c^	10.67 ± 0.12 ^d^	21.34 ± 0.05 ^b^

Results are tabulated as mean ± SEM (*n* = 8). Statistical analyses were performed using one-way ANOVA; means for the Control same parameter with different letters (a–e) in each column are significantly different (*p* < 0.01), the largest data value take the letter (a) and the smallest data value take the letter (e).

**Table 2 pharmaceutics-13-01937-t002:** Effect of CurNP and ZnONP on hippocampus oxidant and antioxidant biomarkers levels in HFD/STZ-induced and treated rats.

Groups	TBARS (µmoL/mg Protein)	NO (µmoL/mg Protein)	CAT (U/mg Protein)	SOD	GPx	GST	GSH (µmoL/mg Protein)
					(µmoL/min/mg Protien)		
Control	174.29 ± 5.11 ^d^	0.80 ± 0.04 ^c^	0.04 ± 0.00 ^d^	105.63 ± 5.40 ^g^	44.28 ± 0.08 ^a^	40.10 ± 0.10 ^c^	24.31 ± 1.32 ^e^
HFD/STZ	381.07 ± 24.50 ^a^	5.41 ± 0.01 ^a^	0.01 ± 0.00 ^f^	18.89 ± 2.00 ^i^	19.42 ± 0.14 ^c^	10.44 ± 0.30 ^f^	8.06 ± 0.90 ^f^
HFD/STZ-Cur	188.95 ± 5.56 ^d^	0.64 ± 0.02 ^c^	0.04 ± 0.00 ^d^	121.30 ± 11.70 ^f^	38.3 ± 0.09 ^b^	36.22 ± 0.50 ^d^	33.23 ± 0.90 ^d^
HFD/STZ-CurNP-10	148.19 ± 5.94 ^f^	0.52 ± 0.01 ^d^	0.10 ± 0.00 ^a^	733.31 ± 10.15 ^a^	47.58 ± 0.12 ^a^	77.30 ± 0.40 ^a^	71.78 ± 2.35 ^a^
HFD/STZ-CurNP-50	165.54 ± 15.41 ^e^	0.71 ± 0.03 ^c^	0.06 ± 0.00 ^c^	380.48 ± 7.50 ^c^	44.14 ± 0.07 ^a^	65.33 ± 0.30 ^b^	60.18 ± 2.36 ^b^
HFD/STZ-Zinc sulfate	226.20 ± 7.76 ^c^	0.61 ± 0.02 ^c,d^	0.04 ± 0.00 ^d^	84.82 ± 7.88 ^h^	33.26 ± 0.10 ^b^	25.43 ± 0.20 ^e^	36.45 ± 1.70 ^d^
HFD/STZ-ZnONP-10	177.88 ± 5.35 ^d^	0.77 ± 0.04 ^c^	0.06 ± 0.00 ^c^	227.82 ± 7.50 ^d^	40.12 ± 0.05 ^a^	50.43 ± 0.10 ^c^	55.97 ± 1.80 ^b^
HFD/STZ-ZnONP-50	158.05 ± 6.90 ^e^	0.66 ± 0.03 ^c^	0.08 ± 0.00 ^b^	529.68 ± 24.40 ^b^	44.78 ± 0.65 ^a^	60.33 ± 0.50 ^b^	65.00 ± 2.90 ^a^
HFD/STZ-Metformin	250.51 ± 4.00 ^b^	1.27 ± 0.02 ^b^	0.02 ± 0.00 ^e^	160.35 ± 16.32 ^e^	35.68 ± 0.15 ^b^	34.55 ± 0.50 ^d^	47.86 ± 0.90 ^c^

Results are tabulated as mean ± SEM (*n* = 8). Statistical analyses were performed using one-way ANOVA; means for the same parameter with different letters (a–i) in each column are significantly different (*p* < 0.01), the largest data value takes the letter (a) and the smallest data value takes the letter (i).

**Table 3 pharmaceutics-13-01937-t003:** Effect of CurNP and ZnONP on serum adipokines levels and hippocampus inflammatory biomarkers.

Groups	Adiponectin (ng/mL)	Leptin (ng/mL)	IL-6 (ng/mg Protein)	TNF-α (ng/mg Protein)
Control	7.46 ± 0.17 ^b^	15.56 ± 0.08 ^c^	17.45 ± 0.023 ^e^	4.63 ± 0.02 ^e^
HFD/STZ	3.40 ± 0.23 ^d^	29.70 ± 0.15 ^a^	68.45 ± 0.03 ^a^	47.36 ± 0.04 ^a^
HFD/STZ-Cur	5.63 ± 0.22 ^c^	23.56 ± 0.12 ^b^	26.41 ± 0.02 ^d^	15.34 ± 0.02 ^c^
HFD/STZ-CurNP-10	6.63 ± 0.15 ^b^	21.23 ± 0.16 ^b^	22.60 ± 0.02 ^e^	9.43 ± 0.01 ^d^
HFD/STZ-CurNP-50	7.93 ± 0.26 ^b^	16.63 ± 0.25 ^c^	25.71 ± 0.02 ^d^	14.03 ± 0.01 ^c^
HFD/STZ-Zinc sulfate	5.46 ± 0.06 ^c^	25.63 ± 0.17 ^b^	55.34 ± 0.02 ^b^	22.85 ± 0.04 ^b^
HFD/STZ-ZnONP-10	7.10 ± 0.11 ^b^	17.63 ± 0.18 ^c^	31.24 ± 0.03 ^d^	17.53 ± 0.01 ^b^
HFD/STZ-ZnONP-50	9.03 ± 0.19 ^a^	19.00 ± 0.12 ^c^	24.30 ± 0.01 ^d^	10.95 ± 0.04 ^d^
HFD/STZ-Metformin	6.27 ± 0.03 ^b^	22.30 ± 0.17 ^b^	43.11 ± 0.01 ^c^	19.83 ± 0.02 ^b^

Results are tabulated as mean ± SEM (*n* = 8). Statistical analyses were performed using one-way ANOVA; means for the same parameter with different letters (a–e) in each column are significantly different (*p* < 0.01), the largest data value take the letter (a) and the smallest data value take the letter (e).

**Table 4 pharmaceutics-13-01937-t004:** Regulation of Aβ-42 clearance, levels of IDE, and activity of AChE in the hippocampus of HFD/STZ-induced rats after treatment with CurNP and ZnONP.

Groups	AChE (mmoL/min/mg Protein)	IDE (ng/mg Protein)	Hippocampal Aβ-42 (ng/mg Protein)
Control	15.58 ± 0.30 ^b^	30.10 ± 0.05 ^a^	23.43 ± 0.01 ^f^
HFD/STZ	84.01 ± 4.10 ^a^	7.12 ± 0.01 ^c^	140.87 ± 0.34 ^a^
HFD/STZ-Cur	17.37 ± 0.57 ^b^	23.00 ± 0.01 ^b^	52.00 ± 0.00 ^d^
HFD/STZ-CurNP-10	12.16 ± 0.14 ^c^	24.24 ± 0.12 ^a,b^	31.53 ± 0.02 ^e^
HFD/STZ-CurNP-50	14.60 ± 0.47 ^b^	29.16 ± 0.08 ^a^	37.42 ± 0.01 ^e^
HFD/STZ-Zinc sulfate	16.63 ± 0.46 ^b^	17.09 ± 0.00 ^b^	101.33 ± 0.02 ^b^
HFD/STZ-ZnONP-10	17.02 ± 0.49 ^b^	27.17 ± 0.09 ^a^	72.55 ± 0.01 ^c^
HFD/STZ-ZnONP-50	11.30 ± 0.80 ^c^	25.08 ± 0.04 ^a^	33.06 ± 0.03 ^e^
HFD/STZ-Metformin	11.12 ± 0.54 ^c^	22.04 ± 0.02 ^b^	60.03 ± 0.02 ^c^

Results are tabulated as mean ± SEM (*n* = 8). Statistical analyses were performed using one-way ANOVA; means for the same parameter with different letters (a–f) in each column are significantly different (*p* < 0.01), the largest data value takes the letter (a) and the smallest data value takes the letter (f).

## Data Availability

All data are available in the manuscript.

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
