# Peer review of "Protective Effect of Natural Antioxidant, Curcumin Nanoparticles, and Zinc Oxide Nanoparticles against Type 2 Diabetes-Promoted Hippocampal Neurotoxicity in Rats"

_pharmaceutics, 2021, doi:10.3390/pharmaceutics13111937_

Round 1
Reviewer 1 Report
CurNP is a natural phytochemical extracted from turmeric and ZnONP is a metal oxide nanoparticle.
To investigate the effect on type 2 diabetes-related neurological changes, in this study, the drug was orally administered to type 2 diabetes-induced mice treated with a high-fat diet and streptorotocin for 6 weeks to improve behavioral disorders, learning ability, biochemical or Molecular changes were measured.
CurNP and ZnONP at appropriate concentrations improve learning and memory functions, improve oxidative and reductive inflammation, up-regulate Bcl2 expression in the hippocampus, down-regulate phosphorylation levels of the MAPK/ERK pathway, a signaling pathway that regulates proliferation and cell survival, Alzheimer’s and It showed efficacy such as hyperphosphorylation of related tau, reduction of accumulation of amyloid beta causing memory impairment, and significant improvement of histological lesions of the hippocampus.
The results of the study demonstrated the efficacy of CurNP and ZnONP as potential therapeutic protective agents for the treatment of type 2 diabetes-related neurological changes.
- The dosage of CurNP and ZnONP is different. What is the criterion for determining the appropriate amount of each therapy?
- There is not much information on the characterization of CurNP and ZnONP, but it would be nice to have more information to compare the morphological and functional characteristics of the two nanoparticles.
- Antioxidant levels in the brain of each group of mice were measured. Why did you conduct an experiment on oxidative stress?
- Why was pure curcumin treated in the drug treatment group during the mouse experiment?
Author Response
- The dosage of CurNP and ZnONP is different. What is the criterion for determining the appropriate amount of each therapy? We used CurNP and ZnONP at two doses (10 mg/kg and 50 mg/kg) in the current study. The goal of this research was to see how a high dose (50 mg/kg) and a low dose (10 mg/kg) of both nanoparticles affected the induced neurotoxicity in T2DM-induced animals. Both high and low nanoparticle doses were chosen based on previous nanoparticle studies and were less than the LD50 value in rats. Both doses are therefore safe for rats.
- There is not much information on the characterization of CurNP and ZnONP, but it would be nice to have more information to compare the morphological and functional characteristics of the two nanoparticles. Thank you so much for this suggestion. We added the FTIR analysis of both nanoparticles besides TEM, size, and PDI (Figures 1 C & 1D).
- Antioxidant levels in the brain of each group of mice were measured. Why did you conduct an experiment on oxidative stress?
- Oxidative stress is a key pathogenic factor in both neurogenerative and metabolic diseases. The brain is vulnerable to excessive oxidative insults because of its abundant lipid content, high energy requirements, and weak antioxidant capacity, so it is an easy target for excessive oxidative insults. In addition, reactive oxygen species increase susceptibility to neuronal damage and functional deficits, via oxidative changes in the brain in neurodegenerative diseases.
- Generally, oxidative stress is a lack of balance between the formation of reactive oxygen species and the efficiency of enzymatic and nonenzymatic antioxidative systems. This results in oxidative damage to cell components and thus leads to the impairment of cell structures and biological functions. It is assumed that in the brain, the formation of oxidative stress is related to the increase in oxidation of FFA and glucose, as well as increased generation of oxygen free radicals in the mitochondrial respiratory chain. Additionally, the overproduction of reactive oxygen species may be caused by the increased glycolysis under hyperglycemic conditions as well as the intensification of nonenzymatic glycation (glycosylation) of cellular proteins. The resulted in oxidative products, especially advanced glycation end products (AGE), play a key role in cerebral neurodegeneration.
- Unfortunately, there is a lack of data comparing HFD-related oxidative stress in the brain, so one of our goals in this study was to assess redox homeostasis, enzymatic and nonenzymatic brain antioxidants, and oxidative damage in the brains of T2DM-induced obese rats, as well as to investigate the effect of our current treatment options in scavenging oxidative stress and increasing brain antioxidant capacity.
- Why was pure curcumin-treated in the drug treatment group during the mouse experiment? The study from the last three decades showed that curcumin exhibits low intrinsic activity, poor absorption, reduced bioavailability, rapid metabolism, and elimination. Nanotechnology-based drug delivery will probably be a suitable method for increasing the biological action of curcumin, which increases its absorption. So, one of the main objectives of this study is to compare the classical form of curcumin (pure curcumin) and nano-curcumin on different mechanisms to support the rationale of using nanoparticles in treatment and protection.
Misspellings, grammar rules, and sentence structure were all checked throughout the manuscript.
Reviewer 2 Report
The manuscript is well written with sufficient information. It can be accepted after minor spell check.
Author Response
Thank you for your valuable comment. Misspellings, grammar rules, and sentence structure were all checked throughout the manuscript.
Reviewer 3 Report
Manuscript number: pharmaceutics-1440414
Article Type: Article
Title: Protective effect of natural antioxidant, curcumin nanoparticles, and zinc oxide nanoparticles against type 2 diabetes-promoted hippocampal neurotoxicity in rats
The Authors were studied
The Authors have investigated the capability of a natural compound, curcumin nanoparticle (CurNP), and biomedical metal, zinc oxide nanoparticle (ZnONP) to alleviate the hippocampal modifications in T2DM-induced rats. According to authors, CurNP and ZnONP appear to be potential neuroprotective agents to mitigate diabetic complications-associated hippocampal toxicity.
Comments:
Introduction: could You show more references from last 3 years?
Conclusions: I suggest make a shorter sentences because first has 5 lines.
General comment: minor revision
Author Response
Introduction: could You show more references from last 3 years?
Thanks a lot for this addition. We added recent references in the introduction section.
Conclusions: I suggest make a shorter sentences because first has 5 lines.
It was changed as advised and some sentences were rephrased. Thank you for this suggestion.
Round 2
Reviewer 1 Report
accept